# Regulatory T Cells in Ovarian Carcinogenesis and Future Therapeutic Opportunities

**DOI:** 10.3390/cancers14225488

**Published:** 2022-11-08

**Authors:** Emily Cassar, Apriliana E. R. Kartikasari, Magdalena Plebanski

**Affiliations:** School of Health and Biomedical Science, Royal Melbourne Institute of Technology, Bundoora, VIC 3083, Australia

**Keywords:** cancer, regulatory T cells, ovarian cancer, combination therapy, epigenetics, chemotherapy, immunotherapy

## Abstract

**Simple Summary:**

Regulatory T cells work to suppress the response of immune cells. In ovarian cancer, tumors may exploit immune suppression to escape destruction by the immune system. We propose that targeting regulatory T cells with medications could improve ovarian cancer survival by preventing immune suppression.

**Abstract:**

Regulatory T cells (Tregs) have been shown to play a role in the development of solid tumors. A better understanding of the biology of Tregs, immune suppression by Tregs, and how cancer developed with the activity of Tregs has facilitated the development of strategies used to improve immune-based therapy. In ovarian cancer, Tregs have been shown to promote cancer development and resistance at different cancer stages. Understanding the various Treg-mediated immune escape mechanisms provides opportunities to establish specific, efficient, long-lasting anti-tumor immunity. Here, we review the evidence of Treg involvement in various stages of ovarian cancer. We further provide an overview of the current and prospective therapeutic approaches that arise from the modulation of Treg-related tumor immunity at those specific stages. Finally, we propose combination strategies of Treg-related therapies with other anti-tumor therapies to improve clinical efficacy and overcome tumor resistance in ovarian cancer.

## 1. Introduction

Ovarian cancer is the deadliest gynecological cancer, with patients in advanced stages surviving for five years less than 30% of the time. This poor survival stems from late diagnosis, frequent metastasis into the abdomen, and a high recurrence of treatment-resistant disease; furthermore, in 70% of cases, malignancy is not identified until an advanced stage [1,2]. One of the biggest obstacles to treatment is immune escape by cancer cells [3,4]. As an immunogenic type of cancer, such as ovarian cancer, could modulate populations of immune cells within the tumor microenvironment [5]. One method by which ovarian cancer may escape the anti-tumor response is by recruiting suppressive Tregs to the cancerous tissue [4]. In this review, we examine the development and function of Tregs and provide evidence of the link between Tregs and ovarian cancer. Based on this, we present Treg-targeting treatments as an effective pathway to improving ovarian cancer outcomes either alone or in combination with existing or emerging therapies. 

### 1.1. Tregs as an Immune Suppressor

Regulatory T cells (Tregs) are T cells that make up approximately 5% of T cell populations and act to suppress autoimmunity and modulate immune function [6]. Tregs are implicated in cancers as they work in the body to suppress any autoimmune response, and in cancer, this action may reduce anti-tumor immunity by allowing cancerous cells to evade any anti-tumor response [6,7]. The most well-characterized Tregs are CD4^+^FOXP3^+^ and other Treg populations, such as CD8^+^ subsets, have been identified but much of their functions remain uncertain. Having an increased amount of effector CD8^+^ T cells infiltrating a tumor may indicate improved patient survival [6]. Further to this, the ratio of CD4^+^CD25^+^FOXP3^+^ Tregs to effector CD8^+^ T cells in the tumor can be an important predictor of mortality in ovarian cancer, with higher amounts of Tregs compared to the effector CD8^+^ T cells being correlated with poorer clinical outcomes and reduced long-term survival [8]. Two origins give rise to distinct types of Tregs: natural Tregs (nTregs), which develop in the thymus, or peripherally derived Tregs, which are produced when circulating naïve CD4^+^ T cells are exposed to transforming growth factor-β (TGF-β) in the periphery. These induced Tregs (iTregs) may also be generated in vitro from other CD4^+^ T cells [9]. Tregs are distinguished from other CD4^+^ T helper subsets in vitro by distinct surface markers that they express, such as elevated levels of CD25 expression and a low or lack of expression of CD127 [10]. Tregs also express forkhead box P3 (FOXP3) transcription factor, a highly specific intracellular transcription factor essential in the development of Tregs. Although Tregs can be FOXP3^-^, most are FOXP3^+^ [11]. Although there are many subtypes of Tregs expressing their specific surface markers, the typical Treg phenotype is CD4^+^CD25^hi^FOXP3^+^ [12]. These cells, along with the highly suppressive FOXP3^hi^CD45^−^ Tregs are most often identified as being of interest in ovarian cancer as they are highly correlated with worse outcomes [8,13,14].

### 1.2. Tregs in Ovarian Cancer

Tregs migrate into ovarian tumors, which is believed to be due to the action of C-C motif chemokine 22 (CCL22) in the tumor microenvironment. Tregs found in ovarian cancer lesions suppress the immune response to tumor-associated antigens, which is achieved by the suppression of Interferon gamma (IFN-γ) and interleukin-2 (IL-2) secretion by the effector T cells. Due to this suppression, Tregs can be an important indicator of prognosis in ovarian cancer patients [4]. Patients with ovarian tumors who were found to have Tregs in their tumors had a higher death hazard ratio and an increased number of Tregs within tumors was associated with more aggressive forms of cancer [15] (Figure 1).

## 2. Molecular and Cellular Pathways in Treg Development

### 2.1. FOXP3 Controls Development of Tregs That Infiltrate Tumors

FOXP3 is a transcription factor that upregulates the expression of Treg-specific genes by cooperatively binding along with other transcription factors and it is not present in other T cells. FOXP3^+^ Tregs may also infiltrate tumors, contributing to immune escape [19].

FOXP3 is also critical for the fitness of Tregs and is controlled by epigenetic factors [9,20]. This epigenetic control is critical in the differentiation of naïve T cells into Tregs. Areas of the untranslated regions of the *FOXP3* locus are hypomethylated in Tregs while being methylated in other T cells. These important regions are referred to as Treg-specific demethylated regions (TSDRs). In addition, the *FOXP3* locus has a pattern of histone modification that makes it more active in Tregs and inaccessible in other T cells. Histones H3 and H4 are acetylated in Treg *FOXP3*, and histone H3 is also trimethylated, opening the chromatin to transcription factors [20]. Naïve T cells show extensive methylation of an upstream enhancer region of *FOXP3*, while natural Tregs are completely unmethylated. This pattern of methylation can be altered by interleukin-6 (IL-6); in nTregs, methylation increase in this region and inhibit the activity of *FOXP3* [21].

### 2.2. IL-2 Receptor Activity Regulates FOXP3 Expression and Treg Maturation

Interleukin 2 receptor (IL-2R) is present on the surface of Tregs and has a role in maintaining the expression of FOXP3 (Figure 2). An underexpression of IL-2R has been found to reduce the expression of FOXP3 in deficient mice [22]. As such, targeting the IL-2 receptor of Tregs may be an indirect way of targeting FOXP3. Another critical role of IL-2R is to ensure that functional Tregs are generated in the thymus [22]. Maturation of natural Tregs occurs in the thymus in response to IL-2, whereas the periphery IL-2 works to promote the growth of Tregs. Aside from this, IL-2 is not required for the suppressive function of Tregs; instead, it enhances conventional cytotoxic T-cell activity [23,24,25].

### 2.3. TGF-β Induces Treg Growth and Contributes to Immunosuppression

Transforming growth factor β (TGF-β) is a cytokine that induces the expression of FOXP3 in naïve T cells causing them to differentiate into Tregs in the periphery (Figure 2) [26]. Tumors that have recruited Tregs may secrete TGF-β and induce naïve T cells in their surroundings to differentiate into Tregs, furthering the local immunosuppression in the tumor microenvironment [27].

**Figure 2 cancers-14-05488-f002:**
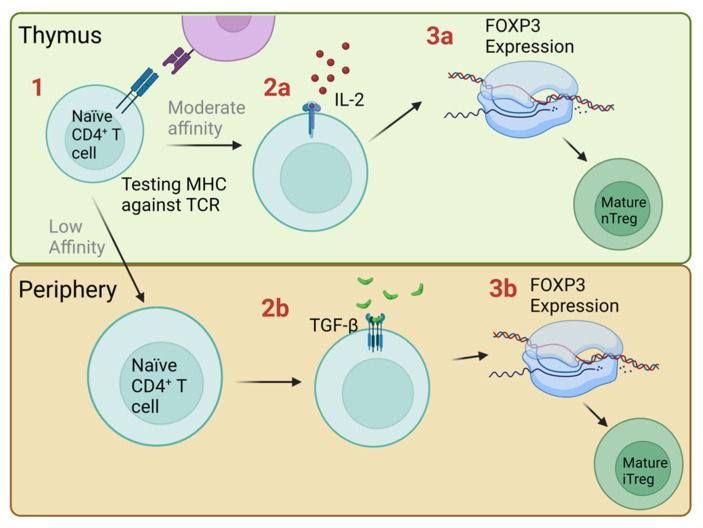
Treg development in the thymus and periphery. (**1**) Naïve CD4^+^ T cells in the thymus test their T cell receptor (TCR) against the major histocompatibility complex (MHC); those with a low binding affinity that are CD25^+^ may circulate in the periphery; others lacking CD25 become helper T cells. Those cells with moderate affinity to self-antigens are selected to become natural Tregs (nTreg). Cells with strong binding to self-antigens undergo apoptosis [28]. (**2a**) Those cells with moderate binding are exposed to IL-2 secreted by thymic cells, which (**3a**) stimulates FOXP3 expression and differentiation into a mature natural Treg [22,29]. (**2b**,**3b**) In the periphery, circulating naïve cells may be exposed to TGF-β, which encourages differentiation into a mature induced Treg (iTreg) [26].

## 3. Molecular and Cellular Pathways That Allow Treg Function

### 3.1. Granzymes Directly Target Other Cells for Apoptosis

Tregs may directly target and destroy other immune cells via the secretion of granzymes, which induce apoptosis in the target cells [30]. In cancer, granzymes A and B are secreted by Tregs. These factors have a dual function: they can destroy immune cells that target cancerous tissue to contribute to immune escape or they can target cancerous cells and destroy them [30,31]. It seems to be the case that Tregs promote immune evasion rather than assist in anti-cancer immunity but their action may be redirected to assist in clearing cancerous cells [31].

### 3.2. PD-1 and PD-L1 Activate Tregs and Suppress Conventional T Cells in Cancer for Immune Escape

Treg cells associated with ovarian tumors have been found to highly express PD-1 while circulating Tregs in those patients with expressed PD-1 at lower levels [32]. PD-1 promotes tolerance of self-antigens by the immune system, suggesting that cancer cells upregulate this protein in Tregs to avoid an immune response. Additionally, PD-1 promotes apoptosis in inflammatory T cells and reduces apoptosis in Tregs [32]. The resulting high concentration of Tregs compared to other T cells may explain why ovarian cancer can be resistant to immunotherapy, as the response of T cells is greatly reduced in the tumor [32]. Expression of programmed cell death ligand 1 (PD-L1) on tissues protects them from autoimmune damage. Some tumors express PD-L1 on their surface, which grants them similar protection from an immune response. Blockade of PD-L1 has been found to induce autoimmunity in mice [33]. Metastatic tumors in the peritoneum have also been identified as having greater expression of PD-L1 than the originating ovarian tumors, although the reason for this is unknown [34].

### 3.3. TNFR2^+^ Tregs Cause Strong Immunosuppression in the Tumor Microenvironment and Ascites

Tumor necrosis factor receptor 2 (TNFR2) is a surface receptor found in the population of Tregs with the highest suppressive activity. Expression of TNFR2 on Tregs is elevated in ovarian cancer patients, possibly due to increased interleukin 6 (IL-6) levels found in the cancer microenvironment [35]. These highly suppressive cells also flock to tumors and ascites due to their high CCR4 expression levels [36]. Activation of TNFR2 promotes cell growth and controls the balance of proliferation and cell death [37]. The expression of FOXP3 can be modulated by TNF in TNFR2^+^ Tregs, leading to a more stable suppressive phenotype [38].

### 3.4. CTLA-4 Mediates Immune Suppression by Tumor-Associated Tregs

Tregs use CTLA-4 to suppress conventional T cells via surface contact. The action of CTLA-4 is to bind CD80 and CD86 competitively with stimulatory signals on antigen-presenting cells to inhibit their activity [39]. The altered function of Tregs is believed to cause several autoinflammatory conditions, with mutations of *CTLA-4* being identified in conditions, such as rheumatoid arthritis, type 1 diabetes, and Graves’ disease [40]. Increased levels of *CTLA-4* expression on Tregs have been found to cause a highly suppressive phenotype [41].

## 4. Treg in Ovarian Cancer Development, Progression, and Resistance

In ovarian cancer, Tregs are implicated in several stages of development (Figure 3). The role of Tregs in oncogenesis is unclear. In later stages of ovarian cancer, Tregs facilitate growth and metastasis of cancer and may contribute to treatment resistance [42,43,44,45]. 

### 4.1. Tregs in Ovarian Oncogenesis

Tumors often begin their growth in an inflammatory environment, making the anti-inflammatory action of Tregs beneficial to the patient [48]. On the other hand, Tregs in the vicinity of a tumor suppress the immune response to cancerous cells [6]. Although it is unknown if Tregs support tumors during early growth in ovarian cancer, Tregs that have been exposed to tumor antigens have been found to strongly reduce the immune response of effector cells in breast cancer tumors [49].

Conversely, in breast cancer, Tregs can influence the transition from carcinoma in situ to invasive cancer [45]. In this case, Tregs have a protective effect on the patient by working to reduce the proliferative ability of cancer cells and the ablation of Tregs can accelerate the progression to the invasive tumor [45]. As both ovarian cancer and breast cancer are solid tumors the protective effect of Tregs may be present in both; however, it is not yet reliably examinable in early-stage ovarian cancer due to difficulties in early detection. 

### 4.2. Tregs Control Inflammation Levels in the Tumor Microenvironment

Tregs may infiltrate into tumors, which is believed to be due to the action of the CCR4 receptor present on Tregs, which binds to the CCL22 ligand produced by tumor cells. Tregs found in ovarian cancer tumors and ascites suppress an immune response to cancer-associated antigens, which is achieved by the suppression of IFN-γ and IL-2 secretion by effector T cells [4]. This immunosuppressive microenvironment supports tumor growth. The infiltration of Tregs into a tumor can reduce the amount or response of cytotoxic cells in the tumor microenvironment and allow cancerous cells to escape destruction by the immune system [50]. Other key factors guiding migration are other chemokine receptors, such as chemokine receptor 6 (CCR6), and integrins, such as lymphocyte function-associated antigen 1 (LFA-1) and very late antigen-4 (VLA-4) [4,51]. The CCR6 receptor detects the appropriate chemokine, CCL20, and causes Tregs to migrate to the source of the attractant. The expression of CCR6 is increased in other cancers, such as liver and colorectal cancer, and has a role in autoimmunity in inflammatory bowel disease [52,53]. The highly suppressive FOXP3^hi^CD45^−^ subset of Tregs are often tumor-infiltrating in solid tumors, such as those found in ovarian cancer. Higher levels of these highly suppressive Tregs in tumors are associated with a worse prognosis [14]. 

Majorly contributing to immune suppression in the tumor microenvironment is TGF-β. In ovarian cancer, tumors may secrete TGF-β, as well as recruit Tregs that contribute to TGF-β levels. Exposure to TGF-β can cause naïve T cells to differentiate into Tregs by inducing the expression of FOXP3 [54]. In the tumor microenvironment, TGF-β also contributes to immunosuppression by reducing the activity of cytotoxic T cells [55].

Tregs may secrete IL-10, which is conventionally thought to contribute to immune suppression in the tumor microenvironment; however, it has also been identified as a factor that limits tumor growth [50,56]. Tumor-associated macrophages also secrete IL-10, which maintains the increased Treg population in the tumor microenvironment [57].

Inflammation has a paradoxical role in tumor formation. Chronic inflammation is thought to be supportive of tumor growth, with inflammatory conditions, such as inflammatory bowel disease being found to progress to malignancy. Some tumors also create an inflamed microenvironment to have ideal conditions to grow [58]. Conversely, immunosuppression caused by Tregs allows the established cancerous cells to have a greater ability to grow and metastasize by reducing anti-tumor immune responses [6]. Treg levels in the blood and tumors of cancer patients have been found to be associated with cancer progression, with those with higher levels of Tregs having a greater chance of fast disease progression and relapse [4,59].

### 4.3. Tregs Promote Ovarian Cancer Angiogenesis

Tregs in and around tumors may also assist in the vascularization of tumors by secreting vascular endothelial growth factor (VEGF) [19,50]. In the tumor microenvironment, VEGF also suppresses the migration of effector immune cells and inhibits the maturation of dendritic cells, which prevents antigen presentation to cytotoxic T cells, contributing to impaired anti-tumor immunity [44,60]. In addition, interferon-γ (IFN-γ) secreted by helper T cells may inhibit angiogenesis and, as such, the reduction in IFN-γ associated with an increased population of Tregs reduces barriers to angiogenesis in cancer [61].

Increased levels of TGF-β also assist cancerous tumors in vascularization [16]. In vitro testing has revealed that tumors show increased angiogenesis when supplemented with TGF-β in a breast cancer model [62]. Thus, in tumors, TGF-β may also play a role in blood vessel integrity. In a mouse model, deficiency of TGF-β during development results in improper angiogenesis and fragile vessels [63].

### 4.4. Tregs Can Assist in Ovarian Cancer Metastasis

Ovarian cancer is known to metastasize into the abdomen via the lymphatic system and rarely in the blood. These new tumors are often resistant to treatment, even more so than the original tumor [43]. Tregs may facilitate tumor metastasis in ovarian cancer both through their suppressive activity and the action of their secreted factors. 

Metastasis can be aided by the presence of VEGF secreted by Tregs in the tumor microenvironment as it increases the permeability of vascular tissue and increases the migration of epithelial cells [44]. Newly formed capillaries that a tumor develops in response to VEGF are often highly permeable to cancer cells and allow increased mobility to other areas of the body. Once cells are in place to metastasize, VEGF also assists in preparing a niche for the cancer cells to grow [46].

In ovarian cancer, increased CCR6 expression in cancer cells has been found to predict poor prognosis. The raised levels of CCR6 promote epithelial-mesenchymal transition of cancer cells, leading to increased cell mobility and metastasis as well as the shedding of cancer cells into the abdominal cavity [18]. The expression of CCR6 is increased in the presence of IL-10, which is secreted by Tregs, and reduced by IFN-γ, which is secreted by the immune cells that Tregs suppress [64]. Interleukin-8 (IL-8) is also secreted by Tregs and has been identified as a promoter of tumor growth and angiogenesis. Elevated levels of IL-8 also activate LFA-1 activity, promoting cell mobility and adhesion [47]. Targeting these chemokines may be an approach to reduce immunosuppression and limit tumor size to improve treatment outcomes.

In ovarian cancer, TGF-β can improve angiogenesis in the tumor and increase infiltration and metastasis of cancerous cells by induction of the epithelial-mesenchymal transition. This occurs due to a decreased growth inhibition TGF-β response in cancer cells while there is increased TGF-β in the tumor microenvironment [17].

### 4.5. Contribution of Tregs to Treatment Resistance in Ovarian Cancer

Ovarian cancer frequently progresses to a treatment-resistant disease [1]. Tregs associated with ovarian cancer are implicated in treatment resistance, usually in reference to treatments that aim to stimulate anti-tumor immunity. In these cases, the suppressive activity of Tregs prevents an appropriate response to the tumor even when encouraged with treatment [65]. Successful treatment with agents, such as cyclophosphamide, is more likely where Tregs have been depleted [42]. Tregs may also be able to act directly on cancerous ovarian cells. In colorectal cancer, Tregs have been shown to increase chemotherapy resistance via secreted factors. Supernatant from Tregs increases the expression of resistance-associated genes in cancer cells [66].

### 4.6. Ratios of Tregs to CD8^+^ T Cells Alter Ovarian Tumor Immunity

Rare CD8^+^CD28^−^ Tregs can transiently produce pro-inflammatory cytokines, such as IFN-γ, IL-17, and IL-10, and aid an immune response [10,11,67]. Although the infiltration of tumors by Tregs generally causes a reduction in the immune response to a tumor, infiltration of CD8^+^ Treg cells has been shown to inhibit naïve CD4^+^ T cell proliferation via TGF-β and IFN-γ secretion. Reducing the population of naïve T cells, which may be induced to differentiate into Tregs, may prevent the over-accumulation of conventional CD4^+^ Tregs in the tumor microenvironment and improve tumor immunity [68].

## 5. Treg-Based Anti-Cancer Therapies

Targeting recurring cancer with drugs that are effective against resistant cells is becoming an increasing challenge. Using a drug that targets Tregs may avoid the issue of resistance by not directly treating resistant tumors, with the reduction in the number of suppressive Tregs impeding the ability of the cancer to grow and spread. Treating Tregs may also improve the patient’s response to other treatments by increasing the immune response to the tumor. The ratio of Tregs to CD8^+^ effector cells is an important prognostic marker in ovarian cancer, with excess Tregs predicting a poorer outcome for the patient [69]. Depletion of Treg populations or disruption of their suppressive function can lead to tumor shrinkage as the immune system is able to respond to cancer cells with anti-tumor immunity [70].

Targeting Tregs may present a more effective way of improving patient survival compared to treating cancerous tissue alone. There are many ways in which Tregs can be targeted by an anti-cancer therapy; these include modification of Treg growth and metabolism including preventing proliferation and promoting depletion of Treg populations, improving anti-tumor immunity with immune checkpoint inhibitors, and altering Treg function by targeting surface receptors and secreted factors (Figure 4) [71].

### 5.1. Immunotherapies Disrupt Signaling between Tregs and Cancer Cells

Immunotherapies can be a versatile way of modulating Treg function during cancer treatment. They may function alone or in combination with other therapies.

Targeting the IL-2 receptor with immunotherapy may allow a more potent immune response to a tumor. Reduced exposure to IL-2 during T-cell development leads to increased autoimmunity in mice. Blocking of the IL-2 receptor of Tregs or anti-IL2 therapy could lead to improved sensitivity to tumor antigens in conventional T cells by reducing Treg activity [74]. An anti-IL2 antibody, F5111.2, has been shown to selectively reduce Treg populations in humans and represents potential future immunotherapy [75]. Targeting IL-2 further reduces the levels of CD8^+^ effector cells in the tumor microenvironment and, in fact, IL-2 injections have been found to increase survival in human trials [76].

Targeting TGF-β with immunotherapy may be a way of preventing cancer progression in situations where malignancy is identified early. This kind of treatment may reduce the population of naïve T cells that are converted to Tregs by TGF-β secreted by the tumor [26]. Treatment with anti-TGF-β therapy for this purpose can reduce the number of Tregs in the ovarian cancer microenvironment and therefore the overall immune suppression resulting from high levels of Tregs. It may also work to inhibit tumor growth and angiogenesis and reduce metastasis to other regions in the body. Care must be taken to ensure that targeting TGF-β does not introduce unwanted systemic effects as it is secreted widely by many cell types and has the potential to be tumor suppressing [77]. Drugs that target the TGF-β pathway already exist, such as vactosertib for colorectal cancer; however, optimal treatment regimens are yet to be developed. In ovarian cancer, targeting TGF-β along with a checkpoint inhibitor is being trialed to develop an appropriate dosage recommendation and identify the ideal patient profile [78].

Targeting immune checkpoints may assist in sensitizing the immune system to cancerous cells by targeting checkpoint molecules, such as programmed cell death protein 1 (PD-1) on exhausted effector T cells. This strategy has already been applied as a cancer immunotherapy. However, checkpoint inhibitors are not currently approved for ovarian cancer due to an elevated risk of relapse [72]. The reason for this is thought to be due to a weaker anti-tumor immune response owing to immunosuppressive cytokines, such as TGF-β and T cell dysfunction in the tumor microenvironment and selection pressure, leading to altered immune checkpoint expression on cancerous cells [6,79]. Although the exhausted effector T cells can be targeted by immune checkpoint inhibitors to regain their function, the limited effectiveness of these drugs could be due to the inhibition of PD-1 also found on Tregs in the tumor microenvironment, which leads to a higher chance of suppression of the effector immune cells by these Tregs [80,81]. It may also be the case that the reduction in PD-1 on the surface of Tregs can lead to a more suppressive phenotype in terms of cytokine secretion or other signaling pathways [33]. Furthermore, ovarian cancer generally has a low mutational burden, reducing the amount of neoantigens present on the cancer cell surface that could provoke an immune response, meaning that a checkpoint inhibitor may never be effective enough to activate the immune response [82]. 

Another immune checkpoint, CTLA-4, can also be targeted by monoclonal antibody treatments and other physical inhibitors. Indeed, the anti-CTLA-4 antibody ipilimumab is approved to treat non-small cell lung cancer and melanoma [83].

Targeting TNFR2 for ovarian cancer treatment is an attractive idea—knockout mice have been found to have increased immune responses to cancer but do not develop systemic autoimmunity [37]. TNFR2 is also highly specific to immunosuppressive cells aside from aberrant expression on the surface of cancer cells, meaning immunotherapies may have less undesirable side effects when compared to treatments with checkpoint inhibitors targeting proteins, such as PD-1 [81].

### 5.2. Depletion of Treg Population and Inhibition of Suppressive Activity

Glucocorticoid-induced TNFR-related protein (GITR) is a costimulatory molecule that is expressed at high levels in Tregs [84]. It is a tumor necrosis factor (TNF) receptor that has previously been implicated in autoimmunity when depleted [85]. Stimulating GITR on both Tregs and effector cells can increase anti-tumor immunity. Activation of GITR in Tregs lessens their suppressive activity and promotes differentiation into helper T cells. In effector T cells, activating GITR deters apoptosis and promotes proliferation as well as signaling to T cells to enter their active state. A GITR agonist, such as a currently trialed immunotherapy, may be useful in ovarian cancer to improve immune responses to tumors [73].

The population of Tregs that has the strongest immunosuppressive activity is TNFR2^+^ [36]. Targeting the TNFR2 protein on Tregs has been found to reduce the proliferative ability of ovarian cancer cells as well as deplete the population of TNFR2^+^ Tregs [37].

### 5.3. Targeting Key Genes for Treg Disruption

Epigenetic drugs (“epidrugs”) can target both writers and erasers of epigenetic marks and are becoming more commonly used in cancer treatment along with other drugs, such as EZH2 inhibitor tazemetostat, which was approved for use in epithelioid sarcoma, and DNMT inhibitor azacytidine, which was approved for leukemia [86]. Treating ovarian cancer with epigenetics may be a replacement for more toxic drugs. Epidrugs would allow the targeting of Tregs-specific factors even where they are not accessible to drugs on the surface of the cells, by directly targeting DNA or RNA. The *FOXP3* gene is a useful target for epidrugs as it is Treg-specific and unstable expression of FOXP3 can compromise the suppressive function of Tregs [87]. Other factors that Tregs rely on for suppressive activity require FOXP3 to maintain expression levels, such as cytotoxic T-lymphocyte-associated protein 4 (CTLA-4) [88]. Using drugs that reduce expression of FOXP3, such as histone acetyltransferase inhibitors or interfering RNA treatments, could reduce the suppressive activity of affected Tregs to alleviate reduced anti-tumor immunity as well as reduce the number of tumors infiltrating Tregs [89]. Likewise, reduced methylation is present at the *CTLA-4* locus in Tregs. In some cancers, overexpression of CTLA-4 has been found to be caused by promoter hypomethylation, which would increase the suppressive activity of Tregs and in turn reduce cancer immunity [88]. Similar to FOXP3, reducing the expression of CTLA-4 with RNA interference or TET inhibitors may improve ovarian cancer outcomes. 

Another potential way of targeting Tregs is via RNA interference (RNAi) or antisense nucleotides [90]. Treating ovarian cancer with RNAi or nucleotides is a highly targeted way of altering Treg function. These are an emerging treatment that may disrupt transcription or translation of genes necessary for Treg functions [90]. These drugs would likely target similar genes as epigenetic treatment would. 

### 5.4. Combining Treg-Targeting Therapy with Existing Drugs

By targeting Tregs during cancer therapy, the opportunity exists to combine treatments to improve the overall benefit to the patient by reducing doses of more toxic drugs, sensitizing the patient to commonly used drugs, or improving the action of other drugs. Some conventional chemotherapies, such as cyclophosphamide or azacytidine, have been demonstrated to deplete Treg populations and improve anti-tumor immunity. Although they do not directly target Tregs, these types of drugs may further improve outcomes when used in combination with more targeted treatments (Figure 5) [84,91].

Combining immunotherapies with Treg-targeting drugs may work to improve the poor response rate of some treatments. Checkpoint inhibitors are one such immunotherapy that have several undesirable side effects during long-term treatment and that have been less effective in ovarian cancer than in other forms of cancer. Despite this, combination therapies utilizing immune checkpoint inhibitors have been found to reduce the size of ovarian tumors in some patients [72]. Checkpoint inhibitors are being trialed for ovarian cancer in combination with other therapies, such as TGF-β inhibitors in the Vigil pilot study (NCT02725489) [78].

Reducing the immune suppression by Tregs may improve the outcomes of chimeric antigen receptor (CAR)-T cell therapy. This treatment involves modifying the patient’s own T cells to better attack cancerous cells for solid tumors and blood cancers. For some patients, CAR-T does not yield positive results due to suppressed anti-tumor immunity and failure of the modified T cells to infiltrate tumors [95]. Modifying Treg function to improve natural anti-tumor immunity may work in tandem with CAR-T to improve the patient’s outcome. Related to CAR-T in that they utilize the patient’s own immune response are therapeutic cancer vaccines [74,75]. These vaccines act to stimulate CD8^+^ T cells to destroy cancer cells, an action which is hindered by Tregs in the areas surrounding tumors and, as such, would benefit from reduced Treg activity [93,94].

Inhibiting DNA repair mechanisms via poly (ADP-ribose) polymerase (PARP) inhibitors as a treatment for ovarian cancer is being investigated by several clinical trials as both a monotherapy and a combination treatment [92]. These drugs may further improve the anti-tumor immune response when combined with a Treg-targeting treatment by encouraging cytotoxic cells to clear the area of damaged and dying cancer cells, which may then move on to directly respond to living tumor tissue. Trials involving these drugs in combination with checkpoint inhibitors and immune priming with cyclophosphamide are currently in phase II, as in SOLACE2 (ACTRN12618000686202) [96].

In addition to simply targeting cancerous tissue or the patient’s immune system, potential treatment could include treatment of other factors that may modulate cancer growth or any immune response, such as potential viral infection [97]. For instance, human cytomegalovirus (HCMV) has been identified as a potential source of immunomodulatory activity in ovarian cancer, with an altered CD8^+^ T cell to Treg ratio noted and increased immunosuppressive cytokines found in the tumor microenvironment [98]. Treatment of these factors may lead to control of immunosuppression and improved recovery.

## 6. Conclusions

The role of Tregs in ovarian cancer is complex and not fully explored. Targeting Tregs when treating ovarian cancer may lead to improved outcomes and less toxicity than directly treating malignant tissue. This can be achieved either by targeting secreted factors, such as cytokines and chemokines; by targeting surface molecules, such as receptors; or by epigenetic modification of Treg function and development. Care must be taken when utilizing these treatments as they may lead to unwanted systemic side effects or inappropriate drug responses. Locating potential new targets for these types of drugs may go a long way to improving treatment outcomes in ovarian cancer patients. These treatments can then be combined with more traditional methods of treatment to improve the overall efficacy of a course of treatment.

## Figures and Tables

**Figure 1 cancers-14-05488-f001:**
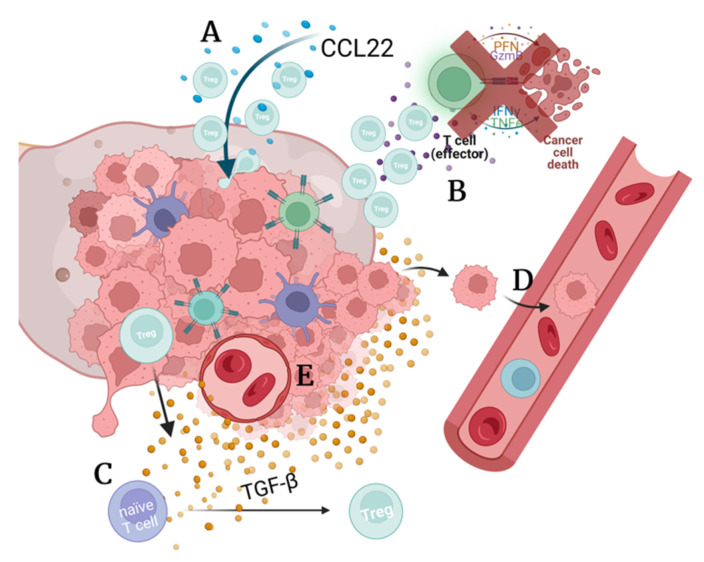
The action of Tregs in the ovarian cancer microenvironment. (**A**) Tumor cells may secrete CCL22 to promote Treg infiltration [4]. (**B**) Tregs in the tumor microenvironment prevent the anti-cancer immunity and allow immune escape by inhibiting the secretion of inflammatory cytokines IFN-γ and IL-2 by effector cells [4]. (**C**) Tumors may secrete TGF-β that aids in differentiating naïve T cells into Tregs, which contribute to immune suppression [16]. (**D**) Metastasis is also increased by TGF-β as it promotes EMT [17]. (**E**) Angiogenesis occurs due to upregulation of TGF-β and IL-8 [17,18].

**Figure 3 cancers-14-05488-f003:**
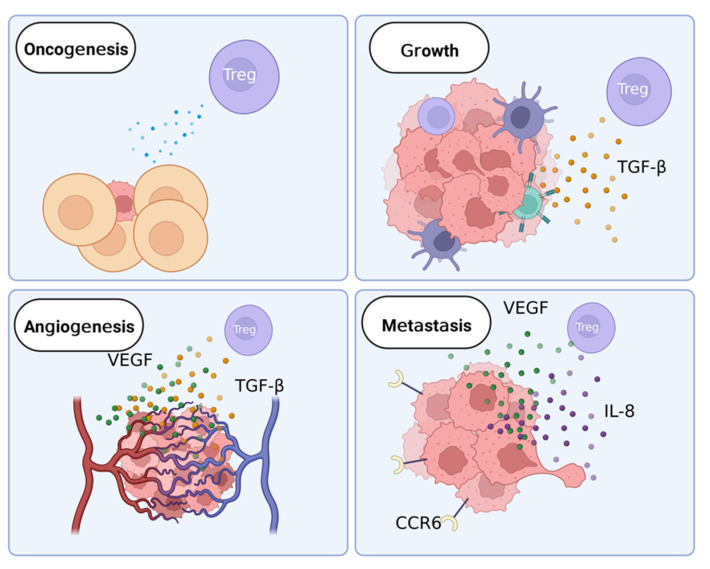
The role of Tregs in cancer development. Tregs may influence early cancer growth via unknown means. In breast cancer, another solid cancer, the presence of Tregs reduces the chance of cancer cells becoming invasive [45]. During cancer growth, tumors recruit immune cells, such as Tregs and tumor-associated macrophages. The main factor secreted by Tregs that assists in proliferation is TGF-β [16]. The secretion of TGF-β encourages growth in cancerous cells and represses anti-tumor immunity. During angiogenesis, TGF-β and VEGF are the primary factors that act on cancer cells [16,44]. These factors are secreted by Tregs and promote the growth of blood vessels and increase the stability of existing blood vessels. In metastasis, VEGF and IL-8 are secreted by Tregs and work to increase cell movement and adhesion [46,47]. Cancer cells also upregulate CCR6 in response to IL-10, which is associated with increased cell mobility [18].

**Figure 4 cancers-14-05488-f004:**
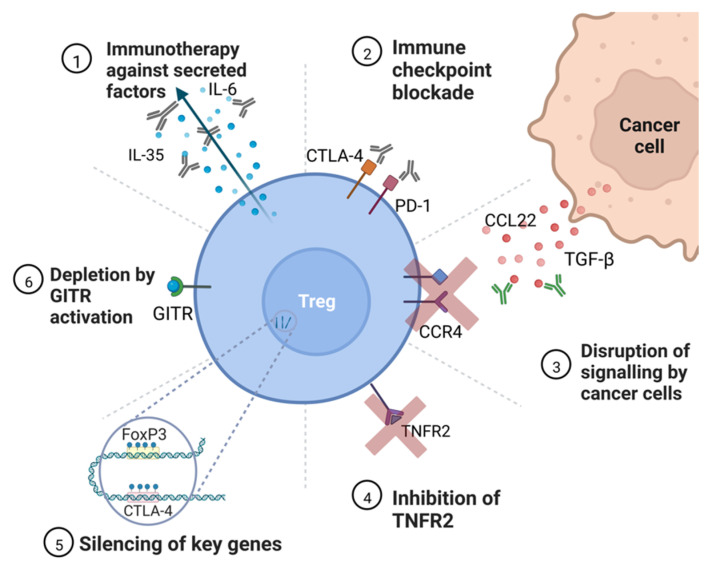
Potential targets for anti-Treg therapy. **➀** Immunotherapy against secreted factors—targeting secreted cytokines may reduce the ability of Tregs to inhibit effector cell activity, notably IL-35, which prevents helper T cell proliferation [64]. **➁** Immune checkpoint blockade—immunotherapy against immune checkpoint molecules, such as CTLA-4 and PD-1, are already in use for some cancers to sensitize the immune system to cancer [72]. **➂** Disruption of signaling by cancer cells—immunotherapies that prevent the action of TGF-β and CCL22 can prevent the accumulation of Tregs in the tumor microenvironment [7]. **➃** Inhibition of TNFR2—TNFR2 is present on the most suppressive Treg populations and a therapy targeting the receptor would deplete Tregs [36]. **➄** Silencing of key genes—silencing of Treg-specific growth genes, such as FOXP3 and CTLA-4, via epigenetic modification or RNAi therapy would prevent the induction of Tregs from naïve T cells and reduce the overall population of Tregs [9,20]. **➅** Depletion by GITR activation—activation of GITR causes Tregs to differentiate into pro-inflammatory helper T cells, with the added benefit of acting on effector cells to improve immunity [73].

**Figure 5 cancers-14-05488-f005:**
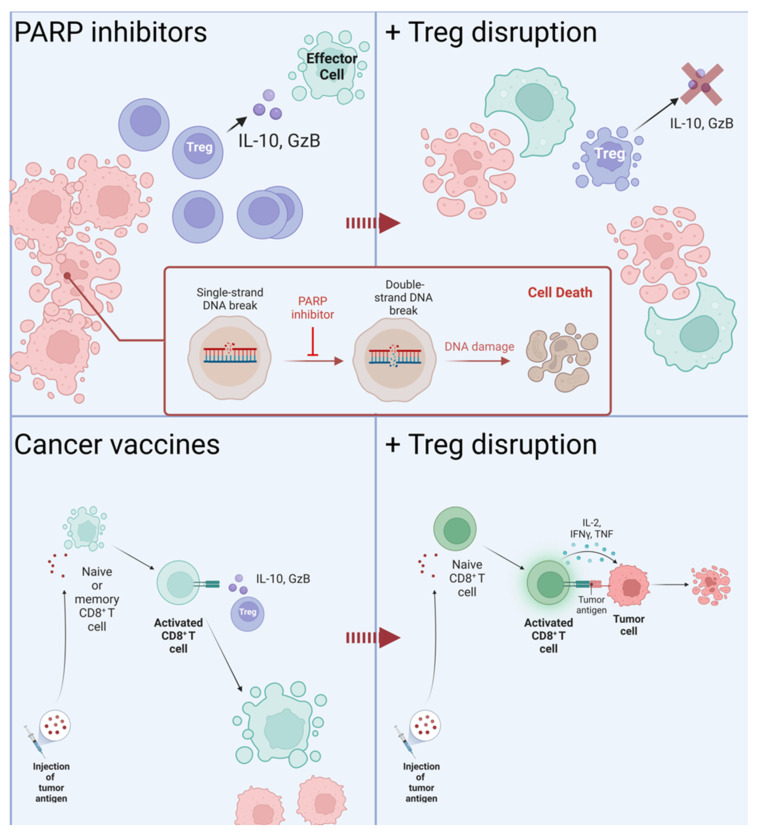
Outcomes of cancer therapy in combination with Treg-targeting treatments. Treatment with PARP inhibitors targets DNA repair mechanisms, causing lethal damage to cancer cells. Apoptosis of cancer cells attracts immune cells to clear cell debris, at which point it may become sensitized to cancer antigens [92]. The presence of Tregs in the tumor microenvironment suppresses the response to cancer antigens; inhibition of their function allows the immune system to begin to respond to cancerous cells. Cancer vaccines increase the immune response to cancer antigens by sensitizing effector cells away from the tumor [93,94]. Without the depletion of Tregs, the benefit of these cells is lost as they are suppressed once they reach the tumor.

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
