# Peer review of "Regulatory T Cells in Ovarian Carcinogenesis and Future Therapeutic Opportunities"

_cancers, 2022, doi:10.3390/cancers14225488_

Round 1

Reviewer 1 Report

I agree with the acceptance of this review paper for publication in the intended article because the roles of regulatory T cells in the carcinogenesis of ovarian cancer as well as potential therapeutic options for patients with ovarian cancer are clearly presented.

Author Response

We believe this review provides practical knowledge and a necessary strategy to improve the clinical outcomes of women with ovarian cancer.

Reviewer 2 Report

This review has a very nice update about ovarian cancer and Tregs potential use in this model as a therapy. Still, I also miss which disadvantages can be present. I consider it necessary to give a general view, good and bad. 

Author Response

We have now added the potential disadvantage of Treg-based therapy, lines 334-340, 345-348, 350-352, and 354-357. As well as in conclusion lines 462-464

Reviewer 3 Report

This review article addresses the role of Tregs in ovarian cancer. Overall, the manuscript reviews major signaling pathways and therapeutic opportunities, and thus adds to the literature. A number of revisions would strengthen the manuscript.

Major:

1.    Section 5.1 – This section would be greatly strengthened with examples of specific drugs against the proposed targets, whether in other tumors, clinical trials, or preclinical work. Such as targets for IL-2, TGF-b signaling, etc.

2.       Line 336-337 – The lack of approval of checkpoint inhibitors in ovarian cancers is more complex than this statement. Please add somewhere in the manuscript a discussion of the low tumor mutational burden of the cancer as well as the lack of efficacy in trials to date.

3.       Section 5.1 – Multiple different targets are all grouped together in the last paragraph of this section. Please restructure.

4.       Please add a reference for the statement in line 136-137.

5.       Section 5.4 – Again, examples of specific therapies would strengthen this section.

Minor:

1.       Line 42 – period should be removed prior to citation

2.       Lines 52-55 – much of this sentence is erroneously duplicated

3.       Lines 71-73, lines 86-89 – same sentence, please delete one

4.       Line 91 – should start with the word “This”

5.       Figure 2 – “Affinity” is misspelled in the figure; also “nTreg” and “iTreg” should be defined in the legend

6.       Line 270 – delete the word “are”

7.       Line 327 – “on” should be “of”

8.       Line 364 – should be “has” and “is”

9.       Line 398 – capitalize “Although”

10.   Line 412 “that have been” – the word “have” is missing

Author Response

Our point to point answer to the reviewer:

Major:

  1. Section 5.1 – This section would be greatly strengthened with examples of specific drugs against the proposed targets, whether in other tumors, clinical trials, or preclinical work. Such as targets for IL-2, TGF-b signaling, etc.

Answer: We have added these suggestions to Section 5.1

  1. Line 336-337 – The lack of approval of checkpoint inhibitors in ovarian cancers is more complex than this statement. Please add somewhere in the manuscript a discussion of the low tumor mutational burden of the cancer as well as the lack of efficacy in trials to date.

Answer: We have also added these suggestions to Section 5.1

  1. Section 5.1 – Multiple different targets are all grouped together in the last paragraph of this section. Please restructure.

Answer: Section 5.1 has now been restructured. The changes are the highlighted sentences.

  1. Please add a reference for the statement in line 136-137.

Answer: A reference has been added

  1. Section 5.4 – Again, examples of specific therapies would strengthen this section.

Answer: We have added the specific therapies into Section 5.4 as highlighted sentences.

Minor:

  1. Line 42 – period should be removed prior to citation
  2. Lines 52-55 – much of this sentence is erroneously duplicated
  3. Lines 71-73, lines 86-89 – same sentence, please delete one
  4. Line 91 – should start with the word “This”
  5. Figure 2 – “Affinity” is misspelled in the figure; also “nTreg” and “iTreg” should be defined in the legend
  6. Line 270 – delete the word “are”
  7. Line 327 – “on” should be “of”
  8. Line 364 – should be “has” and “is”
  9. Line 398 – capitalize “Although”
  10. Line 412 “that have been” – the word “have” is missing

Answers: We have now corrected these minor changes, indicated as highlighted sentences. Figure 2 and its legend have also been corrected.

Round 2

Reviewer 3 Report

Prior comments have been addressed.